# Equivalent MIMO Channel Matrix Sparsification for Enhancement of Sensor Capabilities

**DOI:** 10.3390/s22052041

**Published:** 2022-03-05

**Authors:** Mikhail Bakulin, Vitaly Kreyndelin, Sergei Melnik, Vladimir Sudovtsev, Dmitry Petrov

**Affiliations:** 1Faculty of Radio and Television, Moscow Technical University of Communications and Informatics (MTUCI), 111024 Moscow, Russia; m.g.bakulin@gmail.com (M.B.); vitkrend@gmail.com (V.K.); 2Suisse Department of International Telecommunication Academy, 1202 Geneva, Switzerland; vladimir@soudovtsev.org; 3Department of Information Technology, Peoples’ Friendship University of Russia (RUDN University), 6 Miklukho-Maklaya Str., 117198 Moscow, Russia; petrov-da@rudn.ru

**Keywords:** MIMO, sparse matrix, Turbo coding

## Abstract

One of the development directions of new-generation mobile communications is using multiple-input multiple-output (MIMO) channels with a large number of antennas. This requires the development and utilization of new approaches to signal detection in MIMO channels, since the difference in the energy efficiency and the complexity between the optimal maximum likelihood algorithm and simpler linear algorithms become very large. The goal of the presented study is the development of a method for transforming a MIMO channel into a model based on a sparse matrix with a limited number of non-zero elements in a row. It was shown that the MIMO channel can be represented in the form of a Markov process. Hence, it becomes possible to use simple iterative MIMO demodulation algorithms such as message-passing algorithms (MPAs) and Turbo.

## 1. Introduction

This paper is an extended version of the conference paper [1]. The study introduces and compares various multiple-input multiple-output (MIMO) channel matrix sparsification methods. A description of the Turbo detection algorithm in a MIMO system with a sparsed equivalent channel matrix based on a minimum mean square error (MMSE) detector is also considered. Finally, a comparison of the link-level performances for all of the analyzed algorithms is provided.

One of the directions of the development of new generations of mobile communications is the use of MIMO channels with a large number of antennas. The MIMO technology is one of the most efficient technologies, providing a significant increase in the throughput and an increase in the number of active users. Moreover, the performance grows linearly with an increase in the number of transmitting and receiving antennas. This addresses the requirements for communication systems of 5G and 6G generations [2,3,4].

However, an increase in the number of antennas leads to a significant complication of signal-processing algorithms in MIMO systems. There are many options for constructing MIMO detectors, which differ both in their characteristics and in the complexity of implementation [5,6]. Among the variety of options for constructing MIMO detectors, two limiting cases can be distinguished [7]. At one extreme is the optimal maximum likelihood (ML) receiver, which has the best characteristics but is also the most difficult to implement. Its complexity grows exponentially with the number of antennas and modulation order. At the other extreme, there is a linear MMSE receiver, which has a simple implementation (the complexity grows in proportion to the third power of the number of antennas), but it significantly loses the characteristics to an ML receiver.

In [5], a comprehensive overview of MIMO detectors is given, of which the characteristics are between those of ML and MMSE receivers. There are two directions for improving MIMO detectors: simplifying the ML detector with losses in the energy efficiency and complicating the MMSE detector by increasing the energy efficiency. The first direction includes algorithms based on spherical decoding, e.g., K-best [6,8,9]. The second direction includes various iterative algorithms using the MMSE detector as a basis, e.g., ESA, EPA, and V-BLAST detector [2,10,11,12,13,14,15,16].

It should be noted that a similar problem also exists in the implementation of multiuser detectors in communication systems with multiple access, especially in systems of non-orthogonal multiple access (NOMA) [17,18,19]. However, unlike MIMO systems, communication systems with NOMA have the ability to select various templates and user signals matching a specific type of processing at the receiver. This makes it possible to provide good energy performance with acceptable implementation complexity.

One of the effective approaches of signal detection is based on iterative methods, such as low-density signature (LDS) [20] or sparse code multiple access (SCMA) [3,20,21,22,23,24,25,26,27]. A necessary condition for their use is the ability to represent the channel model in the form of a sparse matrix, as a result of which an individual observation is not a superposition of all symbols at once but only a subset of the symbols. In this case, the number of combinations for one observation will be significantly less than the number of combinations for the entire set of symbols. This makes it possible to use the sequential (optimal) processing of each observation and transfer the received information for the processing of the next observation [8,16,19,28,29,30,31,32]. Similar algorithms are used in Markov processes with finite fixed connectivity.

In MIMO systems, in the general case, the channel matrix is completely filled. Therefore, it is impossible to directly use iterative algorithms to detect MIMO channel signals [14,15,25,28]. This paper proposes a new direction in the development of MIMO detectors. It is based on the transformation of the MIMO channel model to a form convenient for using simple algorithms, i.e., the processing algorithm is not adjusted for the channel model, but the channel model is converted to the processing algorithm.

The goal of the paper is the development of a method for transforming a MIMO channel model into a channel with a sparse matrix containing a limited number of non-zero elements in a row or representing a MIMO channel signal in the form of a Markov process. Hence, it becomes possible to use simple iterative MIMO demodulation algorithms (message-passing algorithms (MPAs), Turbo, etc.). The utilization of these algorithms is especially beneficial in low-power devices, such as, sensors, meter, Internet of things (IoT), and reduced-capacity (RedCap) devices.

The rest of the paper is arranged as follows. In the next section, we introduced a MIMO system model and lay down theoretical prerequisites and potential possibilities of an accurate representation of the MIMO channel matrix in the sparse format. Then, in Section 3, we derived several methods of the approximation of the MIMO channel matrix by a sparse matrix. In Section 4, the modeling and verification of the efficiency of the methods introduced above are presented. Finally, we drew the conclusions in Section 5.

In the paper, the following mathematical symbols, parameters, and operators are used:capital letter, e.g., H used for matrices;det: determinant of a matrix;tr: trace of a matrix;chol: Cholesky decomposition of a matrix;$X^{\prime }$: transpose of matrix X;$X^{-1}$: inverse of a matrix X;$\left|h_{m,n}\right|$: modulo of $h_{m,n}$;$V^{\left(ij\right)}$: block-component matrix of matrix *V*;$p_{pr}\left(x\right)$: the prior distribution;$\Lambda_{eq}\left(x\right)$: the equivalent likelihood function.

## 2. Representation of the MIMO Channel Matrix in the Sparse Format

Let there be an observation:(1)Y=HX+η,
where *Y* is an *N*-dimensional complex vector of signal samples at the input of the MIMO detector, which can be considered a vector of output samples of the MIMO channel, *X* is an *M*-dimensional complex vector of transmitted QAM symbols, which can be considered a vector of input samples of a MIMO channel, *η* is the *N*-dimensional vector of complex samples of the observation noise with zero mathematical expectations and a correlation matrix, $R=2\sigma_{\eta }^{2}I_{N}$ is an identity matrix of size (*N* × *N*), and *H* is a (*N* × *M*)-dimensional complex matrix of the MIMO channel, the elements of which are complex Gaussian random variables with zero mathematical expectations and unit variance for Rayleigh fading. The observation noise variance $\sigma_{\eta }^{2}$ determines the signal-to-noise ratio (*SNR*) at the input of the receiving antenna:$$SNR_{Tx-Rx}=\frac{1}{2\sigma_{\eta }^{2}}.$$

It is known that for any square matrix *H* of full rank, there exists a *QR* decomposition:(2)H=QR,
where *R* is an upper triangular matrix, which is completely described by *N* × (*N* + 1)/2 coefficients, and *Q* is an orthogonal matrix, which is also uniquely determined by *N* × (*N* − 1)/2 coefficients.

From the point of view of the amount of information, such a decomposition does not lead to the appearance of redundancy or to the loss of information, due to the following equation:(3)$$N(N+1)/2+N(N-1)/2=N^{2},$$
which is the amount of information contained in the original matrix *H* that is fully preserved.

Therefore, it can be assumed there exists such a decomposition of the channel matrix:(4)H=QS,
for which matrix *S* contains *N* × (*N* + 1)/2 nonzero elements. The only problem is that these non-zero elements are evenly distributed over all rows, with the average number of non-zero elements in each row less than or equal to (*N* + 1)/2.

If such a transformation shown in Equation (4) is found, then it will allow the use of simpler iterative algorithms for demodulation.

## 3. Methods for the Approximation of the Channel Matrix by a Sparse Matrix

Unfortunately, it is not possible to find the exact decomposition of the channel matrix into an orthogonal and sparse matrix with a fixed number of non-zero entries in a row. The application of numerical methods to test this approach aslo shows that, most likely, such a decomposition does not exist. In addition, even if we assume that such a decomposition exists, then it will allow using iterative demodulation algorithms with a complexity of $N_{it}N2^{MM_{bit}/2}$, instead of an optimal demodulator with a complexity of $2^{MM_{bit}}$, where $M_{bit}=\log_{2}(M)$ is the number of bits in one *M*-QAM symbol. For large *M*, such an algorithm will also be difficult to implement. Therefore, we considered approximate methods for representing a MIMO channel in the form of decimated channels as belows, i.e., such channels in which each observation contains only part of the symbols.

### 3.1. Kullback Distance for the Approximated Channel Model

Let us consider as an approximation criterion the Kullback distance between two distributions, which is determined as follows [33,34]:(5)$$D\left(p,q\right)=\int_{X}p\left(X\right)\ln\left(\frac{p\left(X\right)}{q\left(X\right)}\right)dX,$$
where *p*(*X*) is the original distribution and *q*(*X*) is the approximated distribution.

Let distributions *p*(*X*) and *q*(*X*) be multivariate complex Gaussian distributions and are represented in the following forms:(6)$$\begin{array}{l} q\left(X\right)=\frac{1}{\pi^{M}\det\left(Q\right)}\exp\left\{-{\left(X-\tilde{X}\right)}^{\prime }Q_{}^{-1}\left(X-\tilde{X}\right)\right\}\\ p\left(X\right)=\frac{1}{\pi^{M}\det\left(V\right)}\exp\{-{\left(X-\bar{X}\right)}^{\prime }V_{}^{-1}\left(X-\bar{X}\right)\}, \end{array}$$

Substituting Equation (6) into Equation (5), we can obtain:(7)$$\begin{array}{l} D(p,q)=-\ln\det\left(Q^{-1}V\right)+\\ +\mathrm{tr}\left\{\left(Q^{-1}V\right)-I_{M}\right\}+{\left(\hat{X}-\tilde{X}\right)}^{\prime }Q^{-1}(\hat{X}-\tilde{X}). \end{array}$$

Let us assume we need to approximate a distribution with a fully filled correlation matrix *V* of which the distribution determined by a partially filled correlation matrix *Q* (for example, diagonal, block-diagonal, or sparse) or by a correlation matrix with a certain relationship between elements. In this case, minimizing the distance *D*(*p*,*q*) is reduced for the fulfillment of the equality:(8)$$\tilde{X}=\bar{X},$$
and the minimization of the following functional:(9)$$D(p,q)=-\ln\det\left(Q^{-1}V\right)+\mathrm{tr}\left\{\left(Q^{-1}V\right)-I_{M}\right\},$$
which is subjected to restrictions on the form of the matrix *Q*.

We use the criterion $D(p,q)\underset{\left(\tilde{X},Q\right)}{\to }\min$ for minimizing the distance between the channel matrix *H* by an approximate sparse matrix $\tilde{H}$, i.e., instead of Equation (1), we use the following model:(10)$$\tilde{Y}=\tilde{H}X+\eta ,$$
where matrix $\tilde{H}$ in each row has at most *m* non-zero elements. For now, we consider the case when the placement of these elements in the matrix is given.

As the initial and approximated distributions, we use the posterior distributions obtained for Equations (1) and (10) using the Gaussian prior distribution $X∼N\left({\bar{X}}_{pr},V_{pr}\right)$. The posterior distributions can be found from the Bayes’ theorem:(11)$$\begin{array}{l} p_{ps}\left(X|Y\right)=\frac{L\left(Y|X\right)p_{pr}\left(X\right)}{\int L\left(Y|X\right)p_{pr}\left(X\right)dX}=\\ \frac{1}{\pi^{M}\det\left(V\right)}\exp\{-{\left(X-\hat{X}\right)}^{\prime }V_{}^{-1}\left(X-\hat{X}\right)\}, \end{array}$$
(12)$$\begin{array}{l} q_{ps}\left(X|\tilde{Y}\right)=\frac{L\left(\tilde{Y}|X\right)p_{pr}\left(X\right)}{\int L\left(\tilde{Y}|X\right)p_{pr}\left(X\right)dX}=\\ \frac{1}{\pi^{M}\det\left(Q\right)}\exp\{-{\left(X-\tilde{X}\right)}^{\prime }Q_{}^{-1}\left(X-\tilde{X}\right)\}, \end{array}$$
where L(Y|X) and $L\left(\tilde{Y}|X\right)$ are likelihood functions that are Gaussian according to the models (1) and (10).

It is easy to show that the parameters of the distributions $p_{ps}\left(X\right)$ and $q_{ps}\left(X\right)$ are MMSE solutions (i.e., from the estimates and correlation matrices perspectives) for the corresponding models and are determined by the expressions [7]:

For an accurate model:(13)$$\begin{array}{l} \hat{X}={\bar{X}}_{pr}+K(Y-H{\bar{X}}_{pr})\\ V={\left(H^{\prime }R_{\eta }^{-1}H+V_{pr}^{-1}\right)}^{-1}=V_{pr}-KHV_{pr}\\ K={\left(H^{\prime }R_{\eta }^{-1}H+V_{pr}^{-1}\right)}^{-1}H^{\prime }R_{\eta }^{-1}=V_{pr}H^{\prime }{\left(HV_{pr}H^{\prime }+R_{\eta }\right)}^{-1;} \end{array}$$

For the approximated model:(14)$$\begin{array}{l} \tilde{X}={\bar{X}}_{pr}+\tilde{K}(\tilde{Y}-\tilde{H}{\bar{X}}_{pr})\\ Q={\left(\tilde{H^{\prime }}R_{\eta }^{-1}\tilde{H}+V_{pr}^{-1}\right)}^{-1}=V_{pr}-\tilde{K}\tilde{H}V_{pr}\\ \tilde{K}={\left(\tilde{H^{\prime }}R_{\eta }^{-1}\tilde{H}+V_{pr}^{-1}\right)}^{-1}\tilde{H^{\prime }}R_{\eta }^{-1}=V_{pr}\tilde{H^{\prime }}{\left(\tilde{H}V_{pr}\tilde{H^{\prime }}+R_{\eta }\right)}^{-1.} \end{array}$$

According to Equation (8), the mathematical expectations of the original and approximated distributions should be fulfilled, which are MMSE estimates:(15)$$\hat{X}=\tilde{X}.$$

From Equation (15), we obtain that:(16)$$\begin{array}{l} \tilde{K}\tilde{Y}=\hat{X}\\ \tilde{Y}={\tilde{K}}^{-1}\hat{X}={\left(\tilde{H^{\prime }}\right)}^{-1}\left(\tilde{H^{\prime }}\tilde{H}+2\sigma_{\eta }^{2}I_{M}\right){\left(H^{\prime }H+2\sigma_{\eta }^{2}I_{M}\right)}^{-1}H^{\prime }Y. \end{array}$$

Taking into account that combinations of QAM symbols ${\bar{X}}_{\mathrm{pr}}=0$ and $V_{\mathrm{pr}}=I_{M}$ are independent and equiprobable, as well as the independence of the noise samples $R_{\eta }=2\sigma_{\eta }^{2}I_{N}$, and substituting the expression for the correlation matrix *Q* in Equation (9), we find the conditions which must meet the channel matrix:(17)$$\begin{array}{l} \underset{\tilde{H}}{\tilde{H}=\arg\min D(p,q)}=\underset{\tilde{H}}{\arg\min}\left(-\ln\det Q^{-1}V+\mathrm{tr}\left\{Q^{-1}V-I_{M}\right\}\right)=\\ \underset{\tilde{H}}{\arg\min}\left(-\ln\det\left(\frac{1}{2\sigma_{\eta }^{2}}\tilde{H^{\prime }}\tilde{H}+I_{M}\right)+\mathrm{tr}\left\{\left(\frac{1}{2\sigma_{\eta }^{2}}\tilde{H^{\prime }}\tilde{H}+I_{M}\right)V\right\}\right). \end{array}$$

This condition is minimized under the given constraints on the form of the matrix $\tilde{H}$, i.e., on the arrangement of non-zero elements. Obviously, the minimum value of *D*(*p*,*q*) also depends not only on the value of these elements, but also on their locations.

### 3.2. Diagonal Matrix Approximation

The simplest form of the approximating matrix is the diagonal matrix $\tilde{H}=diag\left\{{\tilde{h}}_{ii},i=\bar{1,N}\right\}$. For it, we obtain:(18)$$\begin{array}{l} \tilde{H}=\\ \underset{\tilde{H}}{\arg\min}\left(-\ln\det\left(\frac{1}{2\sigma_{\eta }^{2}}\tilde{H^{\prime }}\tilde{H}+I_{M}\right)+\mathrm{tr}\left\{\left(\frac{1}{2\sigma_{\eta }^{2}}\tilde{H^{\prime }}\tilde{H}+I_{M}\right)V\right\}\right)=\\ \underset{\tilde{H}}{\arg\min}(-\sum_{m=1}^{M}\ln\left(\frac{1}{2\sigma_{\eta }^{2}}{\left|{\tilde{h}}_{mm}\right|}^{2}+1\right)+\frac{1}{2\sigma_{\eta }^{2}}\sum_{m=1}^{M}{\left|{\tilde{h}}_{mm}\right|}^{2}v_{mm}), \end{array}$$
where $v_{mm}$ are the diagonal elements of the correlation matrix *V*.

Performing differentiation with respect to ${\left|{\tilde{h}}_{mm}\right|}^{2}$ and equating it to zero, we obtain the system of equations:(19)$$-\frac{1}{{\left|{\tilde{h}}_{mm}\right|}^{2}+2\sigma_{\eta }^{2}}+\frac{v_{mm}}{2\sigma_{\eta }^{2}}=0.$$

From here, we obtain the solution:(20)$${\left|{\tilde{h}}_{mm}\right|}^{2}=\frac{2\sigma_{\eta }^{2}}{v_{mm}}\left(1-v_{mm}\right).$$

In this case, the accuracy of the approximation depends only on the modulus ${\left|{\tilde{h}}_{mm}\right|}^{2}$.

### 3.3. Block Diagonal Matrix Approximation

Let the matrix $\tilde{H}$ be represented as a block-diagonal matrix with blocks of size (m×m):(21)$$\tilde{H}=\left[\begin{array}{cccc} {\tilde{H}}^{\left(11\right)} & O_{mm} & \cdots & O_{mm}\\ O_{mm} & {\tilde{H}}^{\left(22\right)} & \cdots & O_{mm}\\ \vdots & \vdots & \ddots & \vdots \\ O_{mm} & O_{mm} & \cdots & {\tilde{H}}^{\left(KK\right)} \end{array}\right],$$
where $O_{mm}$ is a zero matrix of size (m×m).

In this case, the Kullback distance is equal to:(22)$$D=-\ln\det V-\sum_{i=1}^{K}\left(\ln\det\left(\frac{1}{2\sigma_{\eta }^{2}}{\tilde{H}}^{{\left(ii\right)}^{\prime }}{\tilde{H}}^{\left(ii\right)}+I_{m}\right)+\mathrm{tr}\left\{\left(\frac{1}{2\sigma_{\eta }^{2}}{\tilde{H}}^{{\left(ii\right)}^{\prime }}{\tilde{H}}^{\left(ii\right)}+I_{m}\right)V^{\left(ii\right)}-I_{m}\right\}\right),$$
where $V^{\left(ii\right)}$ is the corresponding diagonal block of size (m×m) from the general correlation matrix *V*.

Optimal matrices minimizing this distance are determined by the expressions:(23)$$\left(\frac{1}{2\sigma_{\eta }^{2}}{\tilde{H}}^{{\left(ii\right)}^{\prime }}{\tilde{H}}^{\left(ii\right)}+I_{M}\right)={\left(V^{\left(ii\right)}\right)}^{-1}.$$

From this, we obtain:(24)$${\tilde{H}}^{{\left(ii\right)}^{\prime }}{\tilde{H}}^{\left(ii\right)}=2\sigma_{\eta }^{2}\left({\left(V^{\left(ii\right)}\right)}^{-1}-I_{M}\right).$$

To find the matrix ${\tilde{H}}^{\left(ii\right)}$, one can use known variants of matrix decomposition, for example, SVD decomposition or Cholesky transform [35,36,37,38,39,40]:(25)$${\tilde{H}}^{\left(ii\right)}=\mathrm{chol}\left(2\sigma_{\eta }^{2}\left({\left(V^{\left(ii\right)}\right)}^{-1}-I_{M}\right)\right).$$

In this case, the matrix ${\tilde{H}}^{\left(ii\right)}$ is the upper triangular matrix. As a result of this procedure, we obtain the channel matrix of the following form:(26)$$\tilde{H}=\left[\begin{array}{ccc} \begin{array}{ccc} x & x & x\\ 0 & x & x\\ 0 & 0 & x \end{array} & \begin{array}{ccc} 0 & 0 & 0\\ 0 & 0 & 0\\ 0 & 0 & 0 \end{array} & \cdots \\ \begin{array}{ccc} 0 & 0 & 0\\ 0 & 0 & 0\\ 0 & 0 & 0 \end{array} & \begin{array}{ccc} x & x & x\\ 0 & x & x\\ 0 & 0 & x \end{array} & \cdots \\ \cdots & \cdots & \begin{array}{ccc} x & x & x\\ 0 & x & x\\ 0 & 0 & x \end{array} \end{array}\right].$$

The accuracy of such an approximation is determined by the following value of the Kullback distance:(27)$$D=-\ln\det V+\sum_{i=1}^{K}\ln\det V^{\left(ii\right)}.$$

It can be noted that the value of *D* depends on how the original matrix is split into blocks. Therefore, Equation (27) can be used as an optimization criterion for splitting V into $V^{\left(ii\right)}$ blocks (groups of symbols).

The detection for Equation (10) with a matrix of Equation (26) was carried out separately for each *i*-th block ($i=\bar{1,K}$). The complexity of the optimal demodulation of one such block is proportional to $2^{mM_{bit}}$.

### 3.4. Strip Matrix Approximation

The next version of the sparse matrix representation is a strip matrix, which for *m = 2* has the following form:(28)$$\tilde{H}=\left[\begin{array}{ccccccc} {\tilde{h}}_{11} & {\tilde{h}}_{12} & 0 & \cdots & 0 & 0 & 0\\ 0 & {\tilde{h}}_{22} & {\tilde{h}}_{23} & \cdots & 0 & 0 & 0\\ 0 & 0 & {\tilde{h}}_{33} & \cdots & 0 & 0 & 0\\ \vdots & \vdots & \vdots & \ddots & \vdots & \vdots & \vdots \\ 0 & 0 & 0 & \cdots & {\tilde{h}}_{M-2,M-2} & {\tilde{h}}_{M-2,M-1} & 0\\ 0 & 0 & 0 & \cdots & 0 & {\tilde{h}}_{M-1,M-1} & {\tilde{h}}_{M-1,M}\\ 0 & 0 & 0 & \cdots & 0 & 0 & {\tilde{h}}_{M,M} \end{array}\right].$$

It is known that the correlation matrix can be represented by the Cholesky expansion:(29)$$V=U^{\prime }U,$$
where *U* is the upper triangular matrix. Likewise, $Q^{-1}=G^{\prime }G$.

Then,
(30)$$\begin{array}{l} D=-\ln\det\left(Q^{-1}V\right)+\mathrm{tr}\left\{Q^{-1}V-I\right\}=\\ -\ln\det V-\ln\det\left(G^{\prime }G\right)+\mathrm{tr}\left\{G^{\prime }GV-I\right\}=\\ -\ln\det V-\sum_{i=1}^{M}\ln{\left|g_{ii}\right|}^{2}+\mathrm{tr}\left\{GVG^{\prime }-I\right\}=\\ \ln\det V-\sum_{i=1}^{M}\ln{\left|g_{ii}\right|}^{2}+\mathrm{tr}\left\{GU^{\prime }UG^{\prime }-I\right\}=\\ \ln\det V-\sum_{i=1}^{M}\ln{\left|g_{ii}\right|}^{2}+\sum_{j=1}^{M}\sum_{i=1}^{M}{\left|{\left(ug\right)}_{ji}\right|}^{2}-M, \end{array}$$
where ${\left(ug\right)}_{ij}$ is the *ij*-th element of the matrix $UG^{\prime }$. For a strip matrix G with *m* diagonals, we obtain:(31)$${\left(ug\right)}_{ji}=\sum_{n=0}^{m-1}g_{i,i+n}^{*}u_{j,i+n},\;j\leq i+n.$$

### 3.5. Approximation by a Markov Process

The possibility of using sequential iterative procedures in MIMO detection is allowed by the possibility of representing the evaluated process in the form of a Markov process. In this case, the multivariate posterior distribution is represented as a product of conditional distributions with a finite fixed connection:(32)$$p_{ps}\left(x_{1},\ldots ,x_{M}\right)=\left(\prod_{i=1}^{M-k-1}p_{ps}\left(x_{i}|x_{i+1},\ldots ,x_{i+k}\right)\right)p_{ps}\left(x_{M-k},\ldots ,x_{M}\right),$$
where *k* is the order (memory, connectivity coefficient) of the Markov process.

Having obtained the distribution parameters $p_{\mathrm{ps}}\left(x_{i+1},\ldots ,x_{i+k}\right)$, it is possible to calculate the parameters of the next distribution $p_{\mathrm{ps}}\left(x_{i},\ldots ,x_{i+k-1}\right)$ in Equation (32). In this case, the complexity is determined by the value of the parameter *k*, not by the dimension of the entire vector *M*.

It should be noted that the representation of the channel matrix in the form of a strip matrix with *k* non-zero diagonals also leads to the representation of the posterior distribution in Equation (32). Next, we consider an approach that also follows the MMSE solution, but without using a direct transition to the observation equation.

Let the posterior distribution be obtained as:(33)$$p_{ps}\left(x_{1},x_{2},x_{3},\ldots x_{M}\right)=p_{ps}\left(X/Y\right)=\frac{1}{\pi^{M}\det\left(V\right)}\exp\left\{-{\left(X-\hat{X}\right)}^{\prime }V_{}^{-1}\left(X-\hat{X}\right)\right\}.$$

This distribution can be written in a factorized form:(34)$$p_{ps}\left(x_{1},x_{2},x_{3},\ldots x_{M}\right)=p_{ps}\left(x_{1}|x_{2},x_{3},\ldots x_{M}\right)p_{ps}\left(x_{2}|x_{3},\ldots x_{M}\right)\cdots p_{ps}\left(x_{M-k+1},\ldots x_{M}\right),$$
where
(35)$$\begin{array}{l} p_{ps}\left(x_{i}|x_{i+1},\ldots x_{M}\right)=\frac{p_{ps}\left(x_{i},x_{i+1},\ldots x_{M}\right)}{p_{ps}\left(x_{i+1},\ldots x_{M}\right)}=\\ \frac{1}{\pi v_{i|i+1,\ldots M}^{2}}\exp\left\{-\frac{1}{v_{i|i+1,\ldots M}^{2}}{\left|x_{i}-{\hat{x}}_{i}\left(x_{i+1},\ldots x_{M}\right)\right|}^{2}\right\} \end{array}$$

This conditional distribution can be represented as the result of combining the prior distribution $p_{\mathrm{pr}}\left(x_{i}\right)$ and the equivalent likelihood function:(36)$$p_{ps}\left(x_{i}|x_{i+1},\ldots x_{M}\right)=\frac{\Lambda_{\mathrm{eq},i}\left(x_{i},x_{i+1},\ldots x_{M}\right)p_{pr}\left(x_{i}\right)}{\int_{x_{i}}\Lambda_{\mathrm{eq},i}\left(x_{i},x_{i+1},\ldots x_{M}\right)p_{pr}\left(x_{i}\right)dx_{i}}.$$

Hence, the equivalent likelihood function is determined by the expression:(37)$$\Lambda_{\mathrm{eq},i}\left(x_{i},x_{i+1},\ldots x_{M}\right)=C\frac{p_{ps}\left(x_{i}|x_{i+1},\ldots x_{M}\right)}{p_{pr}\left(x_{i}\right)}.$$

The concept of an equivalent likelihood function, in general, is associated with the concept of external information, which was obtained from general observation and refers only to the considered vector $\left[x_{i},x_{i+1},\ldots x_{M}\right]$ [7].

The idea of representing a sequence $x_{1},x_{2},x_{3},\ldots ,x_{M}$ by a Markov process with a connection of the *k*-th order is to approximate $p_{\mathrm{ps}}\left(x_{i}|x_{i+1},\ldots x_{M}\right)\approx p_{\mathrm{ps}}\left(x_{i}|x_{i+1},\ldots x_{i+k}\right)$ and describe the general distribution in Equation (32). In this case, the equivalent likelihood functions only depend on k + 1 symbols:(38)$$\Lambda_{\mathrm{eq},i}\left(x_{i},x_{i+1},\ldots x_{i+k}\right)=C\frac{p_{ps}\left(x_{i}|x_{i+1},\ldots x_{i+k}\right)}{p_{pr}\left(x_{i}\right)}.$$

For *k* = 0, we obtain a usual approximation by a diagonal matrix.

Taking into account Equation (32), the complete equivalent likelihood function is the product:(39)$$\Lambda_{\mathrm{eq}}\left(X\right)≜C\frac{p_{ps}\left(X|Y\right)}{p_{pr}\left(X\right)}=\left(\prod_{i=1}^{M-k-1}\Lambda_{\mathrm{eq},i}\left(x_{i},x_{i+1},\ldots x_{i+k}\right)\right)\Lambda_{\mathrm{eq},M-k}\left(x_{M-k},\ldots x_{M}\right),$$
where
(40)$$\Lambda_{\mathrm{eq},M-k}\left(x_{M-k},\ldots x_{M}\right)=C\frac{p_{ps}\left(x_{M-k},\ldots ,x_{M}\right)}{p_{pr}\left(x_{M-k}\right)\ldots p_{pr}\left(x_{M}\right)}.$$

Below, we present the rules for calculating the parameters of equivalent likelihood functions.

Firstly, there is an initial distribution:(41)$$p_{ps}\left(x_{1},x_{2},x_{3},\ldots x_{M}\right)=p_{ps}\left(X|Y\right)\;∼\;Ν\left(\bar{X},V\right).$$

From Equation (41), one can select the distribution parameters of the truncated vector:(42)$$p_{ps}\left(x_{i},x_{i+1},\ldots x_{i+k}\right)∼Ν\left({\bar{X}}_{i\ldots i+k},V_{i\ldots i+k}\right).$$

We represent the vector of mathematical expectations and the correlation matrix in the block form:(43)$${\bar{X}}_{i\ldots i+k}=\left[\begin{array}{c} {\bar{x}}_{i}\\ {\bar{X}}_{i+1\ldots i+k} \end{array}\right],$$
(44)$$V_{i\ldots i+k}=\left[\begin{array}{cc} v_{ii} & v_{i,i+1\ldots i+k}^{H}\\ v_{i,i+1\ldots i+k} & V_{i+1\ldots i+k} \end{array}\right].$$

Conditional distribution parameters $p_{\mathrm{ps}}\left(x_{i}|x_{i+1},\ldots x_{i+k}\right)∼N\left({\bar{x}}_{i|i+1\ldots i+k},v_{i|i+1\ldots i+k}\right)$ are determined by expressions:(45)$$\begin{array}{l} {\bar{x}}_{i|i+1\ldots i+k}={\bar{x}}_{i}+T_{i|i+1\ldots i+k}\left(X_{i+1\ldots i+k}-{\bar{X}}_{i+1\ldots i+k}\right)=F_{i|i+1\ldots i+k}{\bar{X}}_{i\ldots i+k}+T_{i|i+1\ldots i+k}X_{i+1\ldots i+k}\\ v_{i|i+1\ldots i+k}=v_{ii}-v_{i,i+1\ldots i+k}^{H}V_{i+1\ldots i+k}^{-1}v_{i,i+1\ldots i+k}=v_{ii}-T_{i|i+1\ldots i+k}v_{i,i+1\ldots i+k} \end{array}$$
where
(46)$$\begin{array}{l} T_{i|i+1\ldots i+k}=v_{i,i+1\ldots i+k}^{H}V_{i+1\ldots i+k}^{-1}\\ F_{i|i+1\ldots i+k}=[\begin{array}{cc} 1 & -T_{i|i+1\ldots i+k} \end{array}]. \end{array}$$

The equivalent likelihood function is determined by the following expression:(47)$$\begin{array}{l} \Lambda_{\mathrm{eq},i}\left(x_{i},x_{i+1},\ldots x_{k}\right)=C\frac{p_{ps}\left(x_{i}|x_{i+1},\ldots x_{k}\right)}{p_{pr}\left(x_{i}\right)}=\\ C\frac{\exp\left(-{\left(x_{i}-{\bar{x}}_{i|i+1\ldots i+k}\right)}^{H}v_{i|i+1\ldots i+k}^{-1}\left(x_{i}-{\bar{x}}_{i|i+1\ldots i+k}\right)\right)}{\exp\left(-x_{i}{}^{H}x_{i}\right)}=\\ =C\exp(-\frac{1}{2\sigma_{\eta }^{2}}{\left|z_{i}-f_{i}X_{i\ldots i+k}\right|}^{2}+{\left|x_{i}\right|}^{2}). \end{array}$$
where
(48)$$\begin{array}{l} z_{i}=f_{i}{\bar{X}}_{i\ldots i+k}\\ f_{i}=F_{i|i+1\ldots i+k}\frac{\sqrt{2\sigma_{\eta }^{2}}}{\sqrt{v_{i|i+1\ldots i+k}^{}}}. \end{array}$$

The equivalent likelihood function $\Lambda_{\mathrm{eq},M-k+1}\left(x_{M-k+1},\ldots x_{M}\right)$ used as the initial one for constructing the demodulation procedure is determined based on its definition—Equation (40)—and has the following form:(49)$$\begin{array}{l} \Lambda_{\mathrm{eq},M-k}\left(x_{M-k},\ldots x_{M}\right)=\\ C\exp\{-{\left(X_{M-k\ldots M}-{\bar{X}}_{M-k\ldots M}\right)}^{H}V_{M-k\ldots M}^{-1}\left(X_{M-k\ldots M}-{\bar{X}}_{M-k\ldots M}\right)+X_{M-k\ldots M}^{H}X_{M-k\ldots M}\}. \end{array}$$

After transformation, this function can be represented as follows:(50)$$\Lambda_{\mathrm{eq},M-k}\left(x_{M-k},\ldots x_{M}\right)=C\exp\left\{-\frac{1}{2\sigma_{\eta }^{2}}{\left\|Z_{M-k}-F_{M-k}X_{M-k}\right\|}^{2}\right\},$$
where
(51)$$\begin{array}{l} Z_{M-k}=F_{M-k}K_{M-k}{\bar{X}}_{M-k\ldots M}\\ F_{M-k}=\mathrm{chol}\left(V_{M-k\ldots M}^{-1}-I_{k+1}\right)\\ K_{M-k}={\left(I_{k+1}-V_{M-k\ldots M}\right)}^{-1} \end{array}$$

The obtained likelihood functions can be used to calculate metrics and implement an iterative demodulation procedure (MPA, Turbo detector, etc.):(52)$$\begin{array}{l} \mu_{i}^{\left(t\right)}=\log\Lambda_{eq,i}\left(X_{i\ldots i+k}^{\left(t\right)}\right)=-\frac{1}{2\sigma_{\eta }^{2}}{\left|z_{i}-f_{i}X_{i\ldots i+k}^{\left(t\right)}\right|}^{2}+{\left|x_{i}^{\left(t\right)}\right|}^{2},\;\mathrm{for}\;i=\bar{1,M-k-1}\\ \mu_{M-k}^{\left(t\right)}=\log\Lambda_{eq,M-k}\left(X_{M-k\ldots M}^{\left(t\right)}\right)=-\frac{1}{2\sigma_{\eta }^{2}}{\left\|Z_{M-k}-F_{M-k}X_{M-k\ldots M}^{\left(t\right)}\right\|}^{2}+{\left\|X_{M-k\ldots M}^{\left(t\right)}\right\|}^{2} \end{array}$$
where $X_{i\ldots i+k}^{\left(t\right)}$ is the *t*-th combination of the (*k* + 1)-dimensional vector $X_{i\ldots i+k}={\left[x_{i}\ldots x_{i+k}\right]}^{T}$, $X_{M-k\ldots M}^{\left(t\right)}$ is the *t*-th combination of the (*k* + 1)-dimensional vector $X_{M-k\ldots M}={\left[x_{M-k}\ldots x_{M}\right]}^{T}$.

It is easy to notice (M−k+1) metrics were calculated with this algorithm, each with a volume $2^{\left(k+1\right)M_{bit}}$. This is significantly less than in the optimal combination algorithm with $2^{MM_{bit}}$ combinations.

Thus, the considered method makes it possible to factorize the likelihood function into factors with a less connectivity, i.e., with a smaller number of observed symbols in individual FP components. This is equivalent to using a decimated channel matrix. Therefore, the methods of the sparsification of the MIMO channel matrix described in the previous sub-sections and the Markov approximation method described in this sub-section solve the same problem of reducing the number of enumerated combinations with the separate processing of the likelihood function components. Such an approach makes it possible to use simple iterative methods.

The considered methods of channel matrix sparsification or FP factorization are approximate, and their accuracy depends on the connectivity coefficient *k* (or on the fill factor of the rows of the sparse channel matrix m/M). However, the accuracy of the approximation also depends on the ordering of symbols during the sparsification procedure, since the considered approaches mainly take into account the correlation of neighboring symbols. Therefore, to improve the accuracy of the approximation, one can preliminarily introduce some ordering of symbols χ=ΠX, where Π is the permutation matrix, which depends on the correlation matrix of the original distribution. Obviously, to improve the accuracy of the approximation, before sparsification, it is necessary to arrange the symbols, so that the largest values of the correlation coefficients are concentrated near the main diagonal.

Figure 1 shows the curves of the Kullback distance D(p,q) between the original posterior distribution and the one approximated by the number of non-zero symbols in the rows of the sparsed channel matrix (connectivity coefficient) for the following algorithms:Algorithm with a block-diagonal matrix—“block”;Algorithm with a strip matrix and the calculation of the coefficients of the sparse matrix by the method of stochastic optimization—“Opt.”;Algorithm with approximation by a Markov process—“Markov’s”.These curves are shown for three options of symbols ordering:Without ordering—“without order”;Simple ordering—“simple order”. In this option, the symbol with the highest total power of all mutual correlation coefficients is put in the first place, and the rest are arranged in descending order of the magnitudes of their mutual correlation coefficients with the first symbols;Serial ordering—“serial order”. In this option, the symbol with the highest total power of (*m* − 1) cross-correlation coefficients is selected, and a set $\Omega^{\left(1\right)}$ of (*m* − 1) symbols having the maximum correlation with the first symbol is set. Next, from this set, the second symbol is selected, with the largest total power of (*m −* 1) cross-correlation coefficients, where (*m −* 2) symbols are already specified and are taken from the set $\Omega^{\left(1\right)}$ (except for the first and second selected symbols) and one symbol is selected from the rest, not included in this set. Then, the third symbol is determined according to the same principle, etc.

With the number of non-zero symbols in the string *m* = 1, all algorithms have the same maximum Kullback distance. This case corresponds to a diagonal approximation, and in terms of demodulation characteristics, it corresponds to a conventional *MMSE* receiver. As *m* increases, as expected, the Kullback distance decreases. Moreover, for the stripe approximation of the channel model and approximation by the Markov process, the distance decreases to a greater extent than with the block approximation. Especially, a strong decrease is observed when using symbol ordering. Sequential ordering is most effective for *m* > 2.

Figure 2 shows a generalized block diagram of the MIMO detection algorithm using the proposed Markov approximation approach. It is based on the *MMSE* receiver, which calculates the *MMSE* estimates vector ${\hat{X}}_{MMSE}$ and the correlation matrix $V_{MMSE}$. These parameters, after ordering, are used to calculate the parameters of the equivalent likelihood functions, which in turn are used to calculate the metrics, which are then used in a soft iterative QAM detector. The output of this MIMO detector is soft bit estimates or a log-likelihood ratio (LLR) for each bit.

Next, we consider the application of Turbo processing in the detection of the signal received after the observation model is parsed. As a result of the performed sparsification procedure, the parameters describing the following likelihood functions can be obtained:(53)$$\begin{array}{l} \Lambda_{\mathrm{eq},i}\left(x_{i},x_{i+1},\ldots x_{i+k}\right)=C\exp\left(-\frac{1}{2\sigma_{\eta }^{2}}{\left|z_{i}-f_{i}X_{i\ldots i+k}\right|}^{2}+{\left|x_{i}\right|}^{2}\right),\;i=\bar{1,M-k-1}\\ \Lambda_{\mathrm{eq},M-k}\left(x_{M-k},\ldots x_{M}\right)=C\exp\{-\frac{1}{2\sigma_{\eta }^{2}}{\left\|Z_{M-k}-F_{M-k}X_{M-k}\right\|}^{2}\}, \end{array}$$
where parameters $z_{i},\;f_{i},\;Z_{M-k},\;F_{M-k}$ can be calculated using Equations (48) and (51).

Let each QAM symbol *x_i_* contains information about *m* bits, i.e., it depends on an *m*-dimensional vector of binary symbols *B_i_*. The total number of bits transmitted by the vector *X* is *mM*. Therefore, we can introduce an (*m* × *M*)-dimensional vector of binary symbols *B*. Let us introduce a notation for the set of indices $\Psi_{i}$ of size *m*(*k* + 1), which joins the numbers of the bits included in the vector $X_{i}={\left[x_{i}\;x_{i+1}\;\ldots \;x_{i+k}\right]}^{T}$, $i=\bar{1,M-K}$.

It is easy to see that the likelihood function $\Lambda_{eq,i}$ contains information about the transmitted bits with numbers $n\in \Psi_{i}$. Therefore, by processing this likelihood function, the transmitted bits with numbers $n\in \Psi_{i}$ can be estimated. It should be noted that the sets $\Psi_{i}$ intersect with each other, so one can get several estimates of the same bit. The task of demodulation is to process all likelihood functions and evaluate each bit using all received information. To solve this problem, it is possible use Turbo processing.

Before processing the *i*-th likelihood function, there is an a priori distribution of binary symbols:(54)$$p_{\mathrm{pr},i}\left(B\right)=\prod_{n=1}^{mM}p_{\mathrm{pr},n,i}\left(b_{n}\right)=\prod_{n=1}^{mM}\frac{1}{2}\left(1+b_{n}\nu_{n,i,\mathrm{pr}}\right)=\prod_{n=1}^{mM}\frac{e^{b_{n}\lambda_{n,i,\mathrm{pr}}}}{e^{\lambda_{n,i,\mathrm{pr}}}+e^{-\lambda_{n,i,\mathrm{pr}}}},$$
where $b_{n}=\pm 1$ is the transmitted information bit.

The parameters describing this distribution are determined by the following expressions
(55)$$\nu_{n,i,\mathrm{pr}}=p_{\mathrm{pr},n,i}\left(b_{n}=1\right)-p_{\mathrm{pr},n,i}\left(b_{n}=-1\right)=\tanh\left(\lambda_{n,i,\mathrm{pr}}\right)$$
(56)$$\lambda_{n,i,\mathrm{pr}}=\frac{1}{2}\log\frac{p_{\mathrm{pr},n,i}\left(b_{i}=1\right)}{p_{\mathrm{pr},n,i}\left(b_{i}=-1\right)}=\tanh^{-1}\left(\nu_{n,i,\mathrm{pr}}\right).$$

Notice that the parameter $\nu_{n,i,\mathrm{pr}}$ is the a priori mathematical expectation of the *n*-th bit before processing the *i*-th likelihood function, and the parameter $\lambda_{n,i,\mathrm{pr}}$ is proportional to the logarithm of the ratio of the prior probabilities of the *n*-th bit before processing the *i*-th likelihood function. At the first processing step, the combinations of all bits are equiprobable, therefore, $\nu_{n,1,\mathrm{pr}}=0$.

With this in mind, the posterior distribution is written as:(57)$$p_{\mathrm{ps},i}\left(B\right)=C\Lambda_{eq,i}\left(B\right)p_{\mathrm{pr}}\left(B\right)=C\left(\Lambda_{eq,i}\left(\left\{b_{n\in \Psi_{i}}\right\}\right)\prod_{n\in \Psi_{i}}p_{\mathrm{pr}}\left(b_{n}\right)\right)\prod_{n\notin \Psi_{i}}p_{\mathrm{pr}}\left(b_{n}\right),$$
where $\left\{b_{n\in \Psi_{i}}\right\}$ denotes a set of symbols *b_n_*, with numbers $n\in \Psi_{i}$.

For those bits that do not belong to the set, we can obtain:(58)$$\lambda_{n,i,\mathrm{ps}}=\frac{1}{2}\log\frac{p_{\mathrm{ps},i}\left(b_{n}=1\right)}{p_{\mathrm{ps},i}\left(b_{n}=-1\right)}=\frac{1}{2}\log\frac{p_{\mathrm{pr},i}\left(b_{n}=1\right)}{p_{\mathrm{pr},i}\left(b_{n}=-1\right)}=\lambda_{n,i,\mathrm{pr}},\;i\notin \Psi_{n}.$$

Taking into account Equations (52), (54) and (57), after a series of transformations, we can obtain:(59)$$\lambda_{n,i,\mathrm{ps}}=\{\begin{array}{ccc} \frac{1}{2}\log\frac{\sum_{t\in t_{n+}}\exp\left(\mu_{i}^{\left(t\right)}+\sum_{n\in \Psi_{i}}b_{n}^{\left(t\right)}\lambda_{n,i,\mathrm{pr}}\right)}{\sum_{t\in t_{n-}}\exp\left(\mu_{i}^{\left(t\right)}+\sum_{n\in \Psi_{i}}b_{n}^{\left(t\right)}\lambda_{n,i,\mathrm{pr}}\right)} & \mathrm{for} & n\in \Psi_{i}\\ \lambda_{n,i,\mathrm{pr}} & \mathrm{for} & n\notin \Psi_{i} \end{array},\;i=\bar{1,M-k},$$
where *t_n_*_+_ is the set of numbers of combinations *t* in which the value of the *n*-th bit in the binary bipolar representation is equal to +1 (or equal to 0 in the binary representation); *t_n_*_−_ is the set of numbers of combinations *t* in which the value of the *n*-th bit in the binary bipolar representation is equal to −1 or is equal to 1 in the binary representation.

These parameters of the posterior distribution describe an independent distribution, i.e., it is the same as the input prior distribution. Thus, these distributions can be used as a priori distributions for the next (*i* + 1)-th step:(60)$$\lambda_{n,i+1,\mathrm{pr}}=\lambda_{n,i,\mathrm{ps}}.$$

As a result, we obtain a sequential multi-step procedure for calculating the posterior parameters of the distribution of the full vector of binary bits *B*.

Figure 3 shows a block diagram of one signal-processing cycle using a sequential multi-step procedure for a model with a Markov approximation.

The resulting multi-step signal-processing procedure at each step uses the optimal algorithm for estimating the vector of the observed binary bits. However, when passing to the next step, not the complete posterior distribution is transferred, but only the parameters describing the independent posterior distribution. Therefore, in the multi-step procedure itself, at each step, the approximation of the full posterior distribution by an independent (factorized) one is used. Obviously, this leads to losses. In such cases, iterative Turbo processing allows for the reduction of losses.

The considered algorithm allows closing the loop and implementing the iterative procedure, using the posterior distribution obtained at the last step of the (*l* − 1)-th iteration as the prior distribution at the first step of the *i*-th iteration, i.e.,
(61)$$\lambda_{n,1,\mathrm{pr}}^{\left(l\right)}=\lambda_{n,M-k,\mathrm{ps}}^{\left(l-1\right)},\;l=1,2,\ldots ,$$

However, according to Equation (59), in the a posteriori parameters $\lambda_{n,M-k,\mathrm{ps}}^{\left(l-1\right)}$, and therefore, in the a priori parameters $\lambda_{n,1,\mathrm{pr}}^{\left(l\right)}$, in part, there is already information obtained during the processing of the *i*-th equivalent observation likelihood function at the previous iteration expressed through its parameter:(62)$$\delta \lambda_{n,i}^{\left(l-1\right)}=\lambda_{n,i,\mathrm{ps}}^{\left(l-1\right)}-\lambda_{n,i,\mathrm{pr}}^{\left(l-1\right)}.$$

Therefore, to avoid the duplication and incorrect accumulation of information, it is necessary to exclude the result of processing the *i*-th equivalent likelihood function at the (*l* − 1)-th iteration from the prior distribution at the *i*-th processing step and at the *l*-th iteration. Consequently, we use the following parameters:(63)$$\lambda_{n,i,\mathrm{pr}}^{\left(l\right)}=\lambda_{n-1,i,\mathrm{ps}}^{\left(l\right)}-\delta \lambda_{n,i}^{\left(l-1\right)}.$$

The result is an algorithm with the block diagram shown in Figure 4.

The complexity of the proposed algorithm is determined by the complexity of the MMSE detector, which is proportional to $M^{3}$, and the complexities of Turbo processing and QAM demodulation, which are proportional to $N_{it}M2^{m\left(k+1\right)}$, where M is the number of transmitting antennas, m is the number of bits in the QAM constellation, $N_{it}$ is the number of iterations in the Turbo algorithm, k is the connective parameters of the models. Variables $N_{it}$ and k are the parameters of the algorithm and can be changed during operation to control the complexity and characteristics of the algorithm. For $N_{it}=1$ and k=0, the complexity and characteristics of the algorithm are the same as those of the MMSE. Note that the complexity of the ML algorithm is proportional to $2^{mM}$, which is significantly above the proposed Turbo algorithm with the sparse transformation of the channel model.

Thus, the proposed approach to sparse the MIMO channel matrix using the Markov approximation method allows simplified Turbo processing algorithms for signal demodulation in MIMO systems.

## 4. Modeling and Verification

In order to verify the efficiency of the proposed channel matrix sparsification algorithm, link-level simulations were carried out with various types of MIMO detectors. The frame error rate (FER) is a generally accepted performance characteristic for the analysis of various algorithms used in communication systems. It allows a comparison of energy efficiency with different types of modulation, coding, and processing algorithms.

Figure 5 shows the dependences of the FER on the SNR per bit (Eb/No) for a MIMO channel with a size of 8 × 8. The results were demonstrated with different approaches for the sparsification of the channel matrix (for *m* = 2) and with different types of MIMO detectors. The simulations were carried out using QPSK modulation and Turbo code with a rate of 1/2. There are results for the following options:“Opt.”—optimal soft MIMO demodulator for the exact model (1);“MMSE”—MMSE demodulator for the exact model. It also corresponds to the variant of the channel model approximation by a diagonal matrix (Section 3.1);“Block”—a demodulator using an approximated block-diagonal MIMO channel model (Section 3.2; Equations (21) and (25)) with a block size of 2 × 2, with an optimal demodulator for each block;“Band, Stoh., Opt”—a demodulator using a striped two-diagonal MIMO channel model (Section 3.3; Equation (28)), in which the parameters of the striped channel matrix are calculated by the stochastic optimization method and the optimal demodulator for this model is used;“Markov’s, Turbo det”—a Markov approximation of the channel model (Section 3.4) with connectivity parameters and iterative detection using the method of equivalent likelihood functions (Equations (47) and (50)) and the principle of Turbo processing (two iterations);“Band, Stoh., MPA det”—iterative MPA detector (three iterations) using a two-diagonal striped MIMO channel model (Section 3.3), in which the channel strip matrix parameters are calculated by the stochastic optimization method.

The results from option (4) above are presented in order to determine the potential capabilities of different variants of the channel model approximation without taking into account the complexity of the detector implementation, i.e., without the influence of suboptimal post-processing.

The curves show that the use of the Markov approximation of the iterative algorithm adopting the method of equivalent likelihood functions and the principle of Turbo processing achieves the same performance as the approximation of the channel model by a strip sparse matrix with the optimization of the coefficients by the method of stochastic optimization and using the optimal soft demodulator. Thus, we can conclude that this approach (Markov approximation) and this demodulation method (Turbo processing + equivalent likelihood functions) achieve theoretically possible characteristics for a given value of the connectivity parameter. In this case, the loss in comparison with the optimal detector and the exact channel model is ~0.8 dB. The difference in the energy efficiency between the Markov approximation (or the approximation of the channel model by a strip matrix) and the block-diagonal approximation of the channel model is about 0.5 dB, in favor of the Markov one.

The MPA demodulation method (variant 6) is similar in complexity in comparison to variant 5 but demonstrates losses of about 0.25 dB.

Figure 6 shows similar curves for a coding rate of 3/4. It also demostrates dependencies on different connectivity coefficients of the Markov model (*k* = 1, 2, 3).

For a given coding rate, the difference between the block approximation and the Markov model is about 1 dB in favor of the Markov model. An increase in the connectivity coefficient of the Markov model increases the energy efficiency but, at the same time, leads to a complication of the demodulation algorithm. For *k* = 3, which is equivalent to the number of non-zero elements in the channel matrix *m* = 4 (half of the total number of symbols), the loss of the proposed approach with respect to the optimal demodulation algorithm is 1–1.5 dB.

Figure 7 shows the FER curves for a 16 × 16 MIMO channel, 16QAM modulation, and Turbo coding with a rate of 3/4. The MIMO detection algorithm with Markov approximation (with a connectivity coefficient k of 1) and Turbo processing with different numbers of iterations are demonstrated. It can be observed that the second iteration leads to an improvement in the energy characteristics by only 0.15 dB. The third iteration does not lead to an improvement in the performance. This is largely due to a fairly effective procedure of symbols ordering.

Compared to the MMSE algorithm (diagonal approximation of the model) for a given MIMO channel configuration, modulation, and coding parameters, the gain in the performance is in the range of 8–14 dB.

## 5. Conclusions

In this paper, efficient methods to approximate the MIMO channel model by sparse matrices or with an equivalent Markov process were introduced and thoroughly analyzed. The proposed approach allows the use of iterative detection methods and leverages the complexity of the implementation of the MIMO detector and its energy efficiency.

Based on the results of link-level simulations, it was demonstrated that the utilization of the proposed approach makes it possible to achieve performance characteristics of a MIMO communication system that are close to the theoretical maximum but with a significantly less complexity. For example, the reduction in the complexity compared to the optimal ML receiver is ~60 times, whereas the SNR decrease does not exceed 0.5–1 dB in an 8 × 8 MIMO configuration, QPSK modulation, with two iterations and a connectivity of the approximated Markov model *k* of 1.

The proposed methods are based on the MMSE solution and can be used as the SNR deteriorates. If the MMSE algorithm provides required FER characteristics, further processing can be excluded. However, if the quality of the MMSE detection is not satisfactory (low SNR), then the first- or second-order Markov model is used. In most cases, it is sufficient to use one iteration.

## Figures and Tables

**Figure 1 sensors-22-02041-f001:**
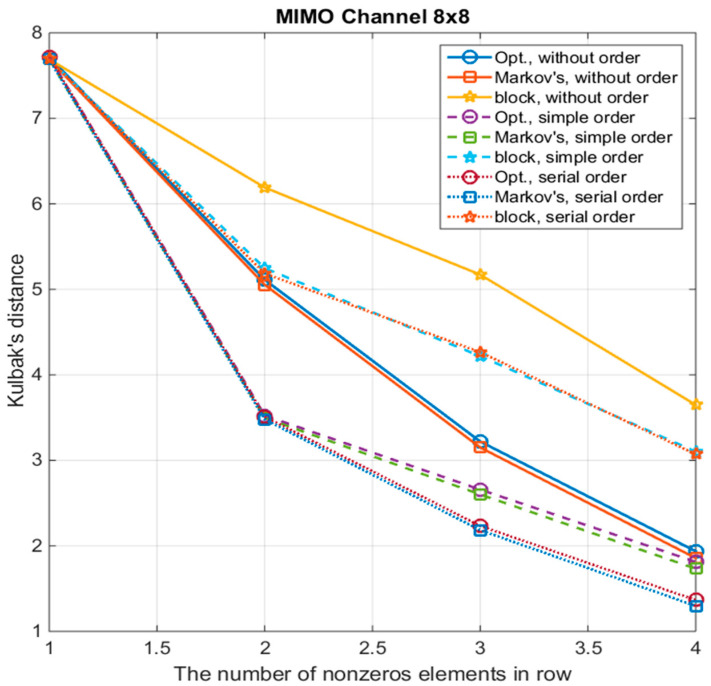
The dependence of the Kullback distance on the number of non-zero symbols in the sparsed channel matrix for different methods of multiple-input multiple-output (MIMO) channel matrix sparsification and for different types of symbol ordering.

**Figure 2 sensors-22-02041-f002:**
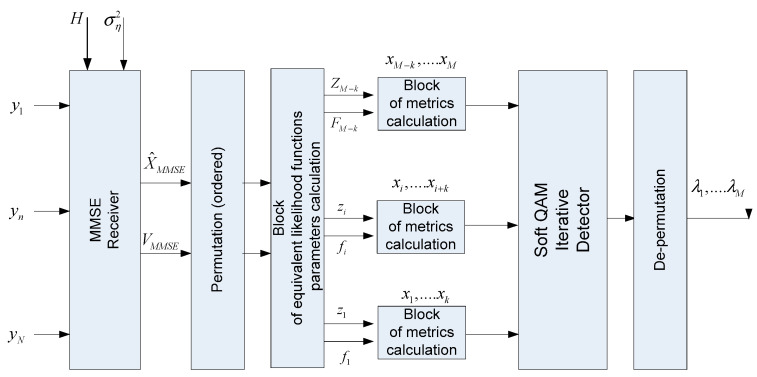
Generalized block diagram of a MIMO demodulation algorithm using an MMSE detector for factorization or MIMO channel matrix sparsification.

**Figure 3 sensors-22-02041-f003:**
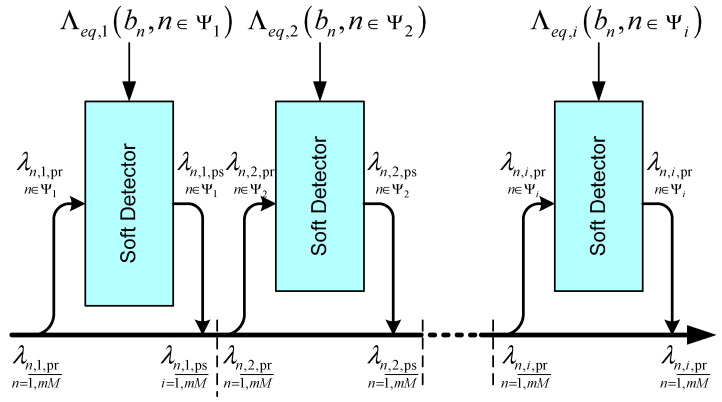
Block diagram of signal processing using a sequential multi-step procedure for a model with Markov approximation.

**Figure 4 sensors-22-02041-f004:**
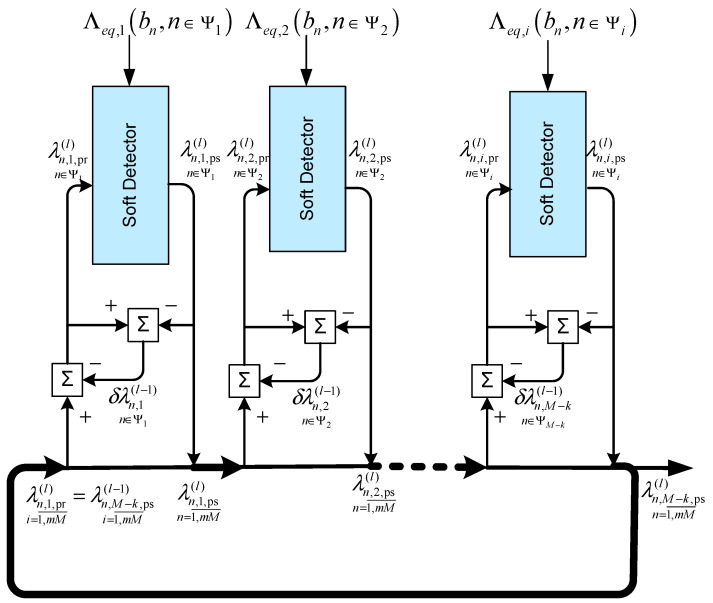
Block diagram of a single-signal Turbo processing using a sequential multi-step procedure for a Markov approximation model.

**Figure 5 sensors-22-02041-f005:**
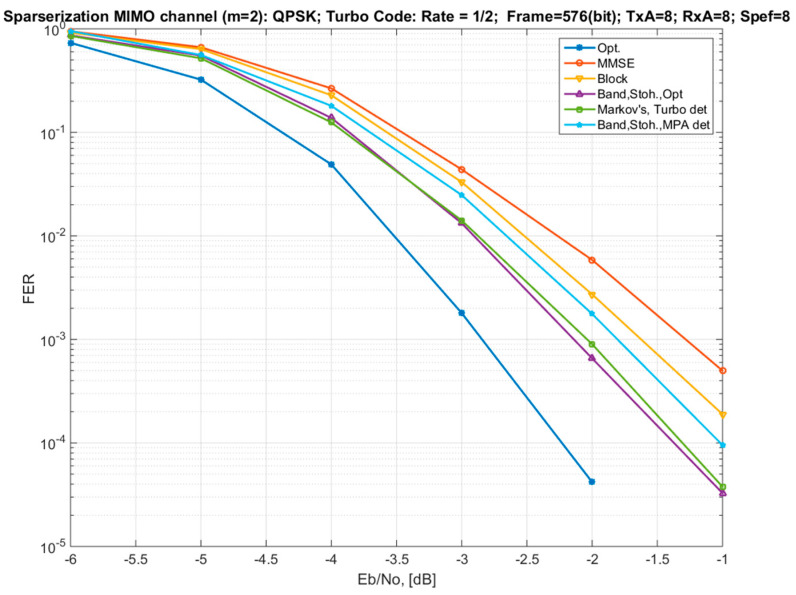
Dependences of the frame error rate (FER) on the signal-to-noise ratio per bit for an 8 × 8 MIMO channel with different options for channel matrix sparsification and for different detectors (QPSK modulation and Turbo code with a rate of 1/2).

**Figure 6 sensors-22-02041-f006:**
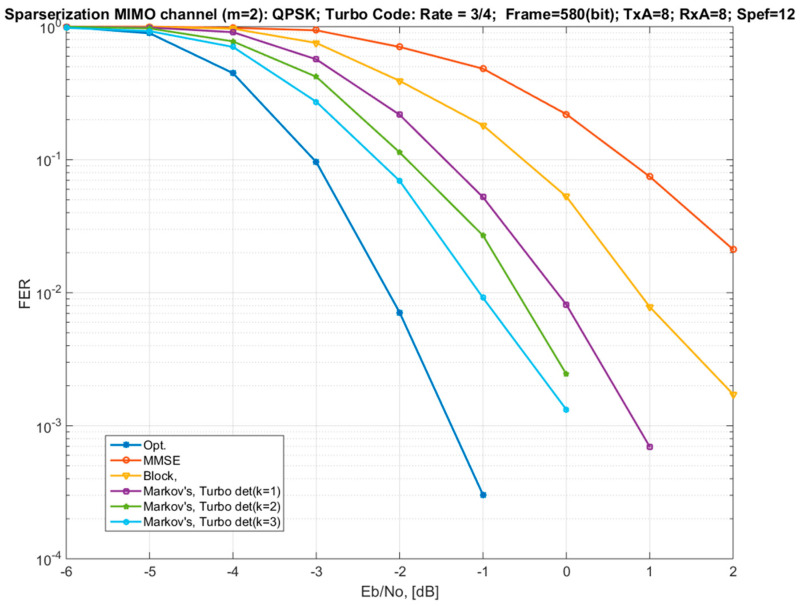
Dependences of the FER on the signal-to-noise ratio per bit for an 8 × 8 MIMO channel with different options for the channel matrix sparsification and with different detectors (QPSK modulation and Turbo code with a rate of 3/4).

**Figure 7 sensors-22-02041-f007:**
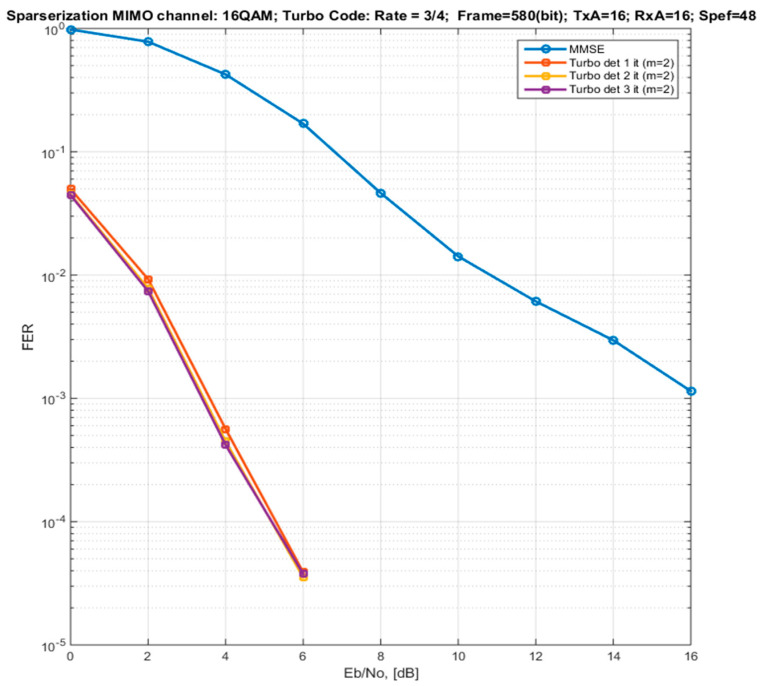
Dependences of the FER on the signal-to-noise ratio per bit for a 16 × 16 MIMO channel for the algorithm with Markov approximation having a different number of iterations and with an MMSE detector (16QAM modulation and Turbo code with a rate of 3/4).

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
