# Peer review of "Equivalent MIMO Channel Matrix Sparsification for Enhancement of Sensor Capabilities"

_sensors, 2022, doi:10.3390/s22052041_

Round 1
Reviewer 1 Report
In this paper, effective sparse matrices or an analogous Markov process approaches for approximating the MIMO channel model are described and fully studied.
The suggested method makes it possible to employ iterative detection methods while also taking use of the MIMO detector's complexity and energy efficiency.
The approaches suggested are based on the MMSE solution and can be employed when the SNR decreases. Further processing can be avoided if the MMSE method offers the appropriate FER features. The first or second-order Markov model is utilised if the quality of the MMSE detection is not adequate (low SNR). In most circumstances, using one that is supported by link-level simulation findings is adequate.
The presented study has to be improved as follows:
1) The introduction is too short and must be extended by including latest relevant publications and also highlight the research gap and the need of the proposed study.
2) The novel contributions should be highlighted at the end of Introdction.
3) Only the FER parameter is evaluted in the experimental results which is insufficient. Various other important parameters should be analysis in numerical simulations to validate the effectiveness. E.g NMSE, SE, EE, Complexity.
4) Conclusions should be re-written to highlight the key performance indicators of the proposed study and also mention the future directions.
5) The paper should undego English Proofreading.
Author Response
We would like to thank the highly respected review for accurately reading our paper and for proving very useful comments!
We did our best to address the comments as follows:
1) The introduction is too short and must be extended by including latest relevant publications and also highlight the research gap and the need of the proposed study.
The introduction was considerably extended to highlight the research gap and the need for the proposed study. We also added a number of publications to the literature review.
2) The novel contributions should be highlighted at the end of Introdction.
A requester clarification was added to the Introduction.
3) Only the FER parameter is evaluted in the experimental results which is insufficient. Various other important parameters should be analysis in numerical simulations to validate the effectiveness. E.g NMSE, SE, EE, Complexity.
We believe that Frame Error Rate (FER) is one of the most used performance characteristics for the analysis of various algorithms used in communication systems. It allows a comparison of energy efficiency with different types of modulation, coding, and processing algorithms. We commented that in the paper.
In addition to that, we have added complexity analysis of the proposed algorithm.
4) Conclusions should be re-written to highlight the key performance indicators of the proposed study and also mention the future directions.
We have extended the conclusion and presented achievable performance gains and complexity reduction benefits.
5) The paper should undego English Proofreading.
We did one more round of English language checks and corrected a number of misprints and simplified some of the sentences.
The changes described above are highlighted in yellow in the new version of the manuscript.

Reviewer 2 Report
The paper Equivalent MIMO Channel Matrix Sparsification for Enhancement of Sensor Capabilities is focused on signal detection in MIMO communication systems. The article describes an innovative method for transforming a MIMO channel into a model based on a sparse matrix with a limited number of non-zero elements in a row. The reason of the technique is to use simple iterative MIMO demodulation algorithms.
The paper is an extended version of an article presented at the conference.
The paper is well organized, it tends towards theoretical methods and techniques rather than practical and experimental. Generally, the language throughout the paper is good, the paper is technically sound, however, I have noticed several typos and minor mistakes, therefore I recommend to perform some minor text editing and corrections.
The adopted methods and presented results are ok.
I recommend to include a list of all mathematical symbols, parameters and operators at the beginning of the paper.
Author Response
We would like to thank the respectful reviewer for accurately reading our paper and for the positive feedback.
As it was recommended, we added a paragraph with the most important mathematical symbols, parameters, and operators at the end of the Introduction. This paragraph is highlighted in green in the uploaded manuscript.
Other changes in the paper are highlighted in yellow.
We also passed the paper through one more round of English language checks. A number of misprints were corrected and a few sentences were improved.

Round 2
Reviewer 1 Report
The authors have addressed my concerns in their revision.
Therefore, the paper is acceptable at this stage.